# Pediatric Multisystem Syndrome Associated with SARS-CoV-2 (MIS-C): The Interplay of Oxidative Stress and Inflammation

**DOI:** 10.3390/ijms232112836

**Published:** 2022-10-25

**Authors:** Serafina Perrone, Laura Cannavò, Sara Manti, Immacolata Rulli, Giuseppe Buonocore, Susanna Maria Roberta Esposito, Eloisa Gitto

**Affiliations:** 1Department of Medicine and Surgery, University of Parma, Viale Antonio Gramsci, 14, 43126 Parma, Italy; susannamariarobertaesposito@unipr.it; 2Department of Human Pathology in Adult and Developmental Age “Gaetano Barresi”, University of Messina, Via Consolare Valeria 1, 98125 Messina, Italy; laura_cannavo@hotmail.it (L.C.); saramanti@hotmail.it (S.M.); immacolatarullo@hotmail.it (I.R.); e.gitto@unime.it (E.G.); 3Department of Molecular and Developmental Medicine, University of Siena, Viale Bracci, 36, 53100 Siena, Italy; giuseppe.buonocore@unisi.it

**Keywords:** oxidative stress, MIS-C, COVID-19

## Abstract

Pediatric inflammatory multisystem syndrome temporally associated with SARS-CoV-2 (MIS-C) is characterized by persistent fever and evidence of single or multiorgan dysfunction, and laboratory evidence of inflammation, elevated neutrophils, reduced lymphocytes, and low albumin. The pathophysiological mechanisms of MIS-C are still unknown. Proinflammatory mediators, including reactive oxygen species and decreased antioxidant enzymes, seems to play a central role. Virus entry activates NOXs and inhibits Nrf-2 antioxidant response inducing free radicals. The biological functions of nonphagocytic NOXs are still under study and appear to include: defense of epithelia, intracellular signaling mechanisms for growth regulation and cell differentiation, and post-translational modifications of proteins. This educational review has the aim of analyzing the newest evidence on the role of oxidative stress (OS) in MIS-C. Only by relating inflammatory mediators to OS evaluation in children following SARS-CoV-2 infection will it be possible to achieve a better understanding of these mechanisms and to reduce long-term morbidity. The link between inflammation and OS is key to developing effective prevention strategies with antioxidants to protect children.

## 1. Introduction

Since April 2020, cases of hyperinflammatory shock have been reported in children with a history of Sars-CoV-2 infection [1]. Symptoms were identical to those of Kawasaki disease, cytokine storm, or toxic shock syndrome, so this new disease was initially named “Kawashocky”, “Coronasacki”, “hyperinflammatory shock in children with COVID-19”, and “pediatric COVID-19 associated inflammatory disorder” [2,3]. Subsequently, the Royal College of Pediatrics and Child Health defined it as “pediatric inflammatory multisystem syndrome temporally associated with SARS-CoV-2” (MIS-C), and the Centers for Disease Control and Prevention and the World Health Organization issued their definitions [1,4].

MIS-C is characterized by persistent fever and evidence of single or multiorgan dysfunction. Dysgeusia, oral pain, exacerbation of autoimmune diseases as well as the herpes simplex virus (HSV) and varicella zoster virus (VZV) infections, lacerations, and aphthous stomatitis were also reported [5]. Laboratory evidence of inflammation, elevated neutrophils, reduced lymphocytes, and low albumin was reported as well [1,4,6].

The median time between primary infection and the incidence of MIS-C symptoms ranges from two to six weeks [7,8].

Although MIS-C shares some aspects with other pathological entities such as toxic shock syndrome (TSS), macrophage activation syndrome (MAS), and Kawasaki disease (KD), the pathophysiological mechanisms of MIS-C are still unknown [9,10].

The common pathway for these life-threatening conditions is that the redox state imbalance reflects oxidative stress during a systemic response to damage [9]. Indeed, since the beginning of 2000, Tsal et al. showed that systemic inflammatory response syndrome (SIRS) was characterized by elevated proinflammatory mediators, including reactive oxygen species (ROS) and decreased antioxidant enzymes [11].

The relationships between ROS generation and inflammation are very complex. ROS may be generated by different mechanisms such as neutrophil and macrophage activation, Fenton chemistry, endothelial cell xanthine oxidase, free fatty acid and prostaglandin metabolism, and hypoxia [12].

The superoxide anion (O_2_^−^), the most abundant radical species, is also the first stage of the bacterial killing reaction, which is followed by production of other free radicals, such as hydrogen peroxide (H_2_O_2_) by superoxide dismutase, hydroxyl radicals catalyzed by transition metals, and HOCl by myeloperoxidase [13]. In a biological system, highly reactive species may generate further unstable molecules such as hydroperoxides and alkoxyl radicals. These substances contribute to bacterial killing but also favor tissue damage. Immunoinflammatory cells may stimulate the endothelium to produce humoral factors and adhesion molecules which in turn attract neutrophils. These events, which are closely linked to macrophage activation, are thought to be implicated in the pathogenesis of MISC-C.

In order to analyze the newest evidence on the role of oxidative stress (OS) in patients diagnosed with MIS-C, a critical review using 2020 PRISMA statement guidelines was conducted. Articles eligible for inclusion were observational cohort, case-control, or randomized controlled trials (RCTs) characterizing serum OS markers in pediatric patients diagnosed with MIS-C (Figure 1). A high sensitivity search strategy was designed combining free text and keyword search term synonym clusters for MIS-C, combined with clusters for cytokines or interleukins.

## 2. MIS-C and Hyperinflammatory Syndromes

Diagnostic criteria for pediatric inflammatory multisystem syndrome temporally associated with SARS-CoV-2 (PIMS-TS) or multisystem inflammatory syndrome (in children) (MIS-C) are reported in Table 1.

MIS-C possesses the symptoms of a number of different disorders, such as KD, toxic shock syndrome, and secondary hemophagocytic lymphohistiocytosis/macrophage activation syndrome. Table 2 shows the peculiar clinical signs of MISC-C and the differential diagnosis with similar diseases.

Therefore, preliminary knowledge of these syndromes can support the description of the pathophysiology in MIS-C, for example, in the systemic inflammatory response syndrome, there is an outperformance of the cellular immune response, similar to what happens in MIS-C patients, in the same way, and the role of oxidative stress is well-known [14]. Several papers reported that in patients with a diagnosis of hyperinflammatory syndrome there are increases in lipoperoxidation, as well as a reduced antioxidant capacity compared to patients without SIRS [11,15,16,17]. Likewise, there is evidence of OS involvement in KD. Some biomarkers of OS, such as reactive oxygen metabolites (ROMs), were increased in patients with KD and decreased after response to treatment [18]. It has also been described that there is a late increase in plasma levels of malondialdehyde and hydroperoxide after acute illness in patients with KD [19]. Furthermore, antioxidant factors are decreased in patients with acute KD [20]. Some studies have proposed the use of antioxidants as adjuvant therapy for these patients.

These assumptions reinforce the hypothesis of a possible role of OS in the pathogenesis of MIS-C. In addition, MIS-C patients have been documented to develop autoantibodies against endothelial cells like KD patients, contributing to the endothelial dysfunction and multisystem inflammation characteristics of these patients [21,22]. ET-1, a 21-amino-acid polypeptide produced by vascular endothelial cells, shows crucial vasoactive properties by binding to specific G protein-coupled membrane receptors, ETA and ETB, respectively expressed on vascular smooth muscle cells and endothelial cells. The ET-1 biosynthesis and release are transcriptionally regulated by several factors such as p38MAP kinase, NF-kB, PKC/ERK, and JNK/c-Jun which in turn are activated in the presence of OS [23]. Increased ET-1 levels may, in turn, promote further production of superoxide radicals resulting in a vicious circle, further contributing to endothelial dysfunction.

Although MIS-C has clinical similarities and cytokine profiles comparable to KD and TSS, recent studies suggest that there are differences in the type of cells activated during the immune response, observing a specific expansion of activated T lymphocytes compared to patients with KD, TSS, or SARS-COV-2 [4]. 

Consiglio C.R. et al. compared the T cell subpopulations between children with KD, MIS-C, and healthy children [24]. This study revealed differences in the distribution of CD4 + T cell subpopulations and in the frequency of T-follicular helper (TFH) cells. Notably, total T cell frequencies were lower in both types of hyperinflammatory patients than in the healthy children. TFH cells were reduced in SARS-CoV-2 infected children, both with and without MIS-C, but not in Kawasaki disease patients, while CD4 + effector T cells were higher in children with MIS-C than in the patients with Kawasaki disease. Differences in T cell subgroups and cytokine mediators place MIS-C at the intersection of Kawasaki disease and immune states of acute SARS-CoV-2 infection in children, as well as hyperinflammatory syndromes. The same authors showed that MIS-C hyperinflammatory states differ from those of the Kawasaki disease. For example, IL-17A mediates hyperinflammation in the Kawasaki disease, but not in MIS-C, so IL-17A blocking agents could be considered in patients with severe Kawasaki disease but not in MIS-C [25,26,27].

NT-proBNP, N-terminal probrain natriuretic peptide; PT, prothrombin; PTT, partial thromboplastin time; KD, Kawasaki disease; CRP, C-reactive protein; ESR, erythrocyte sedimentation rate; IL, interleukin; LDH, lactate dehydrogenase; RT-PCR, reverse transcription-polymerase chain reaction.

## 3. Immune Response in MIS-C

It is known that in inflammatory syndromes there is an uncontrolled and exacerbated immune response. Autoantibodies that bind endothelial cells have been hypothesized to contribute to endothelial dysfunction and the multisystem inflammation typical of MIS-C [21]. Indeed, several authors have demonstrated the presence of IgG and IgA autoantibodies that recognize endothelial antigens in both MIS-C patients and adults with COVID-19 but not in healthy controls [25,28]. To better understand the profile of this anti-SARS-CoV-2 response, Gruber et al. analyzed subclasses of SARS-CoV-2 protein-specific immunoglobulins (anti-La and anti-Jo-1) in children with MIS-C, children and adults with acute SARS-CoV-2 infection requiring hospitalization, and convalescent adults. The plasma of MIS-C patients showed high IgG levels and low IgM levels, as seen in the response of the convalescent patients. However, MIS-C patients showed higher IgA levels compared to convalescent plasma. Indeed, IgA levels in MIS-C patients were similar to those in patients with acute infection [28].

It has been suggested that enhanced oxidative stress (OS) may play a role in the pathogenesis of the disease. In SARS-CoV-2 infection, inflammatory changes may lead to an overproduction of reactive oxygen species (ROS), which could contribute to the progression into severe disease [28,29]. Pincemail et al. showed that critically ill COVID-19 patients had increased lipid peroxidation and deficits in some antioxidants such as vitamin C, glutathione, and thiol proteins [30].

In the same way, MIS-C patients have been shown to also have significant elevations of multiple cytokine families. Dufort et al. demonstrated that 97% of patients with MIS-C had IL-6 ≥ 5.0 pg/mL [31]. Moreover, Gruber et al. showed that especially CCL19, CXCL10, and CDCP1 are involved in the MIS-C pathogenesis, because these citokynes enlist natural killer (NK) and T cells from the circulation and modulate their function [27]. Increased levels of ROS may ultimately cause significant damage to the vascular endothelium. Vella et al. reported that the immunological profile of MIS-C patients was characterized by strong activation of CX3CR1 + CD8 + T lymphocytes of the vascular endothelium, as in severely ill adult patients with COVID-19 [32]. Although many cytokines involved in MIS-C resembled those of acute infection, a specific MIS-C cytokine profile could be distinguished from that of COVID-19. Indeed, Gruber et al. showed that the increase in unique chemokines such as CXCL5, CXCL11, CXCL1, and CXCL6 and cytokines such as IL-17A, CD40, and IL-6 help distinguish MIS-C patients from COVID-19 pediatric patients [27].

## 4. Oxidative Stress in MIS-C

ROS released inside phagocytes during infection and cytokine production is an essential defense mechanism. It also alters the extracellular oxidative balance and harms tissues since the cells undergo an efflux of O_2_

The term ROS includes free radicals, which are atoms or molecules with one or more unpaired electrons. Free radicals may react with other radicals, the unpaired electrons forming a covalent bond. The resulting molecule may decompose other molecules into toxic products. Free radicals may react with nonradical molecules in free radical chain reactions, which are stopped by antioxidant molecules, enzymes, or protein reactions. Superoxide anion [O_2_^−^∙ is the precursor of most ROS and a mediator in oxidative chain reactions. Dismutation of O_2_^−^∙ by superoxide dismutase (SOD) produces H_2_O_2_ which in turn may be fully reduced to water by glutathione peroxidase and catalase or partially reduced to a hydroxyl radical [OH^∙^]. The latter reaction is called the Fenton–Haber Weiss reaction and is catalyzed by reduced transition metals, particularly iron, but also copper and zinc [33]. There is no specific scavenger for this radical and, once released, OH^∙^ reacts with lipoproteins, cell membranes, lipids, proteins, DNA, amino acids, and other molecules causing structural and functional damage to these structures. OH^∙^ is one of the strongest oxidants in nature and may damage tissues. Since the OH^∙^ is formed by the Fenton reaction which is dependent on nonprotein bound iron (NPBI), the conditions of intracellular or extracellular availability of nonprotein bound iron are one of the most important sources of ROS-dependent tissue damage. Superoxide has low activity and poor reactivity. However, it may participate in nitric oxide-mediated reactivity generating oxidative tissue damage. Superoxide and nitric oxide readily react to form an extremely reactive substance called peroxynitrite (ONOO^−^) [33].

The reaction product of NO· and O_2_^−^ is the unstable molecule peroxynitrite (ONOO^−^), which is regarded as highly reactive [33].

The discovery of the homologs of the cytochrome subunit of the NADPH oxidase, the NOX family, demonstrated the importance of ROS generation via NOX since these enzymes have been found in virtually every tissue [34].

Damiano et al. recently reported the redox control hypothesis of host cells in SARS-CoV-2 infection. Virus entry activates NOXs and inhibits Nrf-2 antioxidant response inducing ROS levels [35]. The link between NOXs and ACE2/Ang/Mas receptor axis plays a pivotal role in the pathogenetic mechanisms of COVID-19 and its complications [36]. Accordingly, ACE-2, a homolog of ACE with different substrate specificity that metabolizes angiotensin II into angiotensin, induces the release of ROS and superoxide levels via increases in expression and activation of NADPH oxidases, a major source of superoxide anion in the vasculature. The latter reacts with NO and results in the synthesis of peroxynitrite and endothelial dysfunction.

NOX enzymes modulate fundamental biological processes [37]. While the most relevant generation of ROS by NOX occurs in phagocytes after activation upon inflammatory mediatoris, ROS are produced via NOX by a variety of cell types in response to signals such as formylpeptide receptors (FPRs) and Toll-like receptors (TLRs) [38]. Furthermore, Nox-dependent ROS generation has been suggested to trigger the adaptive response of a variety of stressors [39]. However, NOX-induced ROS generation can activate the NRF2 pathway which increases antioxidant protection during inflammation [40], see Figure 2.

Molecular genetic analyses conducted in clinical studies led to the identification of four polypeptides, gp91phox, p22phox, p47phox, and p67phox, necessary for the good functionality of the respiratory burst. A fifth phox protein, p40phox, exists in a complex with p47phox and p67phox [41] and is important for the production of high levels of superoxide during phagocytosis [42,43].

The following path transfers electrons from NADPH to O_2_ in the respiratory burst:NADPH −330 mV → avin −256 mV → heme −245 mV → O_2_ −160 mV → O_2_^−^

The gp91phox, p22phox, p67phox, and p47phox subunits of the phagocytary respiratory burst oxidase have also been detected in endothelial cells and smooth muscle cells [44]. The latter also show a decrease in gp91phox, p22phox, p67phox, and p47phox subunits and production of agent oxidants. Close homologs of the gp91phox subunit of avocytochrome [45], known as the NOX protein, have also been described. There are six members of NOX in mammalian cells, in addition to gp91phox, which is also designated as NOX2. All NOX proteins share a similar protein structure, with transmembrane domains for heme binding and an avoprotein intracellular domain. The other NOX proteins are expressed in various cell types, including the epithelium, smooth muscle cells, and intestines and seem capable of generating superoxide, albeit at much lower levels than the phagocytic enzyme. Homologs of p67phox (NOXA1) and p47phox (NOXO1) have also been identified which regulate many of the nonphagocytic NOXs. The biological functions of nonphagocytic NOXs are still under study and appear to include: defense of epithelia, intracellular signaling mechanisms for growth regulation and cell differentiation, and post-translational modifications of proteins [46].

Oxidative stress is an important hallmark in several diseases characterized by a predisposition towards alteration of the inflammatory function and/or premature ageing [47]. MIS-C was initially described in children and was much more prevalent in children then in adults (MIS-A). In aged individuals, a pro-oxidant state is also often associated with a mitochondrial dysfunction and a decline in the expression and activity of endoplasmic reticulum molecular chaperones [48]. A delicate balance between oxidants and antioxidants is essential for physiological functioning. On the contrary, the loss of this balance usually leads to dysfunctions and cellular damage at various levels, including membrane phospholipids, proteins, and nucleic acid. Although significantly less data exists for MIS in adults, there is extensive clinical and laboratory overlap between the two conditions [49]. By measuring the redox biomarker profile, such as concentration of protein and lipid peroxidation products, the presence of a pro-oxidant state could be documented, both in children and in the elderly.

This manuscript has some limitations. First, the qualitative synthesis derives from a few studies focusing on the oxidative stress related-pathogenesis of the disease. Second, we could not control for possible environmental factors, and the educational nature of the review leaves the possibility of residual confounding. However, as the condition implies pathways of inflammation, the results of this study offer new potentially useful information for this patient population.

## 5. Conclusions

MIS-C disorder is a systemic hyperinflammation developed by some children following SARS-CoV-2 infection. It shares clinical features and molecular mechanisms with other multisystem inflammatory syndromes, such as KD, TSS, and MAS [50].

While there are differences, these diseases can provide a model for studying the pathogenesis of MIS-C, especially since KD, TSS, and SIRS are thought to be triggered by viral infections and sepsis and can evolve into systemic inflammation and vasculitis.

The deepening knowledge regarding inflammatory and OS mediators in children following SARS-CoV-2 infection would allow to achieve a better understanding of these mechanisms and to reduce long-term morbidity. The link between inflammation and OS is key to develop treatments for effective prevention strategies with antioxidants to protect children.

## Figures and Tables

**Figure 1 ijms-23-12836-f001:**
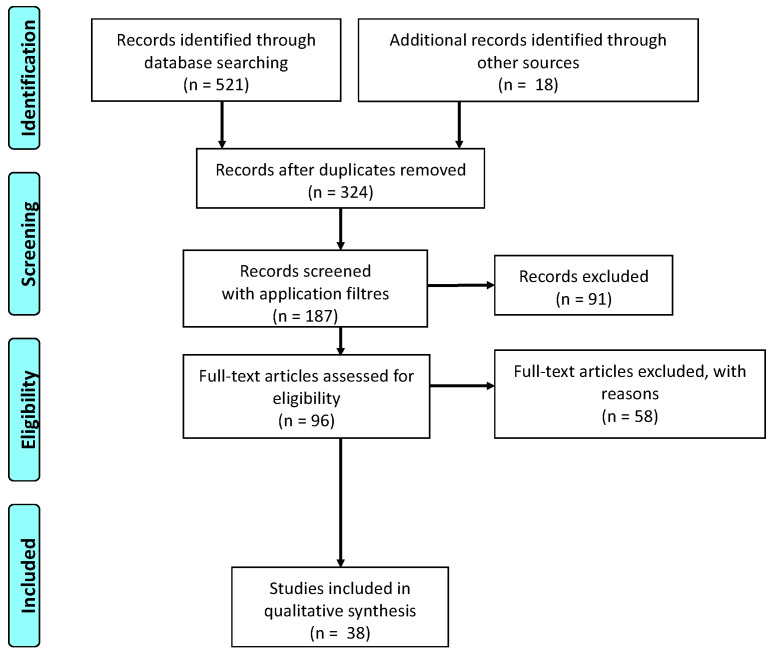
Flow chart of the literature research for two independent reviewers.

**Figure 2 ijms-23-12836-f002:**
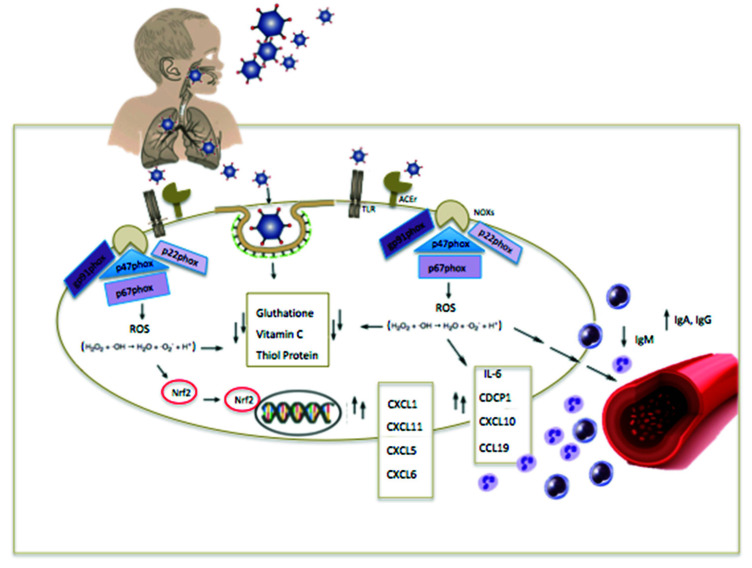
The interplay of oxidative stress and inflammation in MIS-C. SARS-CoV-2 induces pathological responses including increased inflammation, oxidative stress, and endothelial dysfunction, which contribute to progression into severe disease. TLR: Toll-like receptor; ACEr: angiotensin-converting enzyme receptor; ROS: reactive oxygen species; Nrf2: nuclear factor erythroid 2-related factor 2; NOXs: NADPH oxidase enzymes; IL: interleukin; Ig: immunoglobulin.

**Table 1 ijms-23-12836-t001:** Diagnostic criteria for PIMS -TS or MIS-C.

	World Health Organization [4]	Royal College of Pediatrics and Child Health (UK)	Centers for Disease Control and Prevention (US) [1]
**Age**	0–19 years of age.	0–18 years of age.	Individual aged <21 years.
**Clinical feature**	Fever >3 days and 2 of the following: (a) Rash or bilateral nonpurulent conjunctivitis or muco-cutaneous inflammation signs (oral, hands or feet). (b) Hypotension or shock. (c) Features of myocardial dysfunction, pericarditis, valvulitis, or coronary abnormalities (including ECHO findings or elevated Troponin/NT-proBNP), (e) Acute gastrointestinal problems (diarrhea, vomiting, or abdominal pain).	Persistent fever > 38.5 °C. Evidence of single or multiorgan dysfunction (shock, cardiac, respiratory, renal, gastrointestinal or neurological disorder) with additional features, which may include children fulfilling full or partial criteria for KD.	Fever ≥38.0 °C for ≥24 h, or report of subjective fever lasting ≥24 h. Severe illness necessitating hospitalization. 2 or more organ systems affected (e.g., cardiac, renal, respiratory, hematologic, gastrointestinal, dermatologic, and neurological.
**Laboratoristic criteria**	Evidence of coagulopathy (by PT, PTT, elevated D-dimer). Elevated ESR, C-reactive protein, or procalcitonin. No other obvious microbial cause of inflammation, including bacterial sepsis, staphylococcal, or streptococcal shock syndromes.	Neutrophilia, elevated CRP, and lymphopenia	Elevated CRP, ESR, fibrinogen, procalcitonin, D-dimer, ferritin, LDH, or IL-6, elevated neutrophils, reduced lymphocytes, and low albumin.
**COVID-19 relationship**	Evidence of COVID-19 (RT-PCR, antigen test or serology positive), or likely contact with patients with COVID-19.	SARS-CoV-2 PCR testing may be positive or negative.	Positive for current or recent SARS-CoV-2 infection by RT-PCR, serology, or antigen test; or COVID-19 exposure within the 4 weeks prior to onset of symptoms.

**Table 2 ijms-23-12836-t002:** Peculiar clinical signs of MIS-C and the differential diagnosis with similar diseases.

	MIS-C	Kawasaki Disease	Hemophagocytic Lymphohis-Tiocytosis/Macrophage Activation Syndrome (HLH/MAS)	Toxic Shock Syndrome
**Age of affected persons**	4–15 years	<5years		<10 years
**Ethnicity**	Hispanic/Latino/African American	Est Asian	No difference	No difference
**Symptoms**				
**Hypotension**	Present or absent	Generally absent	Generally absent	Present
**Rash**	Generally present	Generally present	Present or absent	Generally present
**Fever**	Present	Present	Present	Present
**Vomiting/Diarrhea/or abdominal pain**	Generally present	Rare	Present or absent	Generally present
**Respiratory distress**	Generally present	Rare	Generally present	Present
**Myocarditis and pericarditis**	Generally present	Rare	Present or absent	Rare
**Hemodynamic instability**	Present or absent	Rare	Present or absent	Present or absent
**Coronary artery dilatation and aneu- rysms**	Present or absent	Present	Generally absent	Generally absent
**Heart involvement**	Generally present	Present	Present or absent	Present or absent
**Mucous Membrane Involvement**	Present or absent	Generally present	Generally present	Rare

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
