# Peer review of "Pediatric Multisystem Syndrome Associated with SARS-CoV-2 (MIS-C): The Interplay of Oxidative Stress and Inflammation"

_ijms, 2022, doi:10.3390/ijms232112836_

Round 1

Reviewer 1 Report

LIne 13: temporarily needs to be changed to temporally

line 26: babies may be changes to children as it is not only babies who gets MIS-C.

line 36: temporarily needs to be changed to temporally

paragraph 64-70: authors has not given what studies were found. it would be appropriate to present the relevant papers on this topic in a table format and summarizing main finding. 

line 83: several data is not accurate use of the word data ! several papers may be used.

line 95-97: please explain how ET-1 pathway is important for oxidative stress in MIS-C patients or how it connects to overall picture. 

line 106: sars-cov-2 experienced children! use of word experience is not appropriate here.  Can be changed to sars-cov-2 infected children or children who had covid19.

line 128: exasperated ! Not sure if author wanted to type exacerbated. it is better to use the word exacerbated.

paragraph 128-140: Author discuss antibody response under ROS which is not directly related to heading (oxidative stress in MIS-C). recommend to move this paragraph to somewhere else in the paper. the same applies to paragraph 147-160. Author may crate new heading as immune response in MIS-C for these 2 paragraphs.

line 134: please explain which viral proteins are targeted by these antibodies 

line 216-217: there is grammatical error and it is hard to understand what the sentence says.  whole sentence needs to be re written.

line 236: Only by is a very strong statement (as if there is no other way). recommend less strong statement. 

line 239-240: the whole sentence needs to be re written. strategies were used twice which makes the sentence confusing and hard to follow. recommend changing the word babies to children 

Figure: recommend adding NRF2 pathway to figure.

ACE1/ACE2 balance is very important as ACE1 is pro inflammatory and pro ROS and ACE2 is anti inflammatory and anti oxidant. Recommend that author talks about the importance of ACE2 pathway and if there is any studies on ACE 2 pathway in MIS-C patients.

is there any studies which looked at the oxidative stress markers with patients with MIS-C, pediatric or adult/MIS-A. 

Author should also notice and discuss this dilemma. Although antioxidant capabilities of the children are much better than elderly MIS-C was initially described in children and  much more prevalent in children then elderly. If oxidative stress is very important in the pathogenesis of MIS-C then why we don't see much higher MIS-C/MIS-A cases in elderly with chronic disease like metabolic syndrome, DM, cardiovascular disease that has common factor of inflammation and elevated oxidative stress.

Author Response

Reviewer 1:

Line 13: temporarily needs to be changed to temporally

Re: the word has been changed

Line 26: babies may be changes to children as it is not only babies who gets MIS-C.

Re: the word has been changed

Line 36: temporarily needs to be changed to temporally

Re: the word has been changed

Line 83: several data is not accurate use of the word data ! several papers may be used.

Re: the word has been changed

line 106: sars-cov-2 experienced children! Use  of word experience is not appropriate here.  Can be changed to sars-cov-2 infected children or children who had covid19.

Re: the word has been changed

line 128: exasperated ! Not sure if author wanted to type exacerbated. it is better to use the word exacerbated.

Re: the word has been changed

line 216-217: there is grammatical error and it is hard to understand what the sentence says.  whole sentence needs to be re written.

Re: the typo and grammar mistakes have been changed. The sentence has been rewritten

line 239-240: the whole sentence needs to be re written. strategies were used twice which makes the sentence confusing and hard to follow. recommend changing the word babies to children 

Re: the sentence has been rewritten

line 236: Only by is a very strong statement (as if there is no other way). recommend less strong statement.

Re: the sentence has been rewritten, with less strong statment

line 239-240: the whole sentence needs to be re written. strategies were used twice which makes the sentence confusing and hard to follow. recommend changing the word babies to children

Re: the sentence has been rewritten, and the word babies has been changed with children

Paragraph 64-70: authors has not given what studies were found. it would be appropriate to present the

relevant papers on this topic in a table format and summarizing main finding. 

Re: A figure with the considered studies has been added (Figure 1)

Line 95-97: please explain how ET-1 pathway is important for oxidative stress in MIS-C patients or how it connects to overall picture. 

Re: the role of ET-1 has been better defined in the text.

paragraph 128-140: Author discuss antibody response under ROS which is not directly related to heading (oxidative stress in MIS-C). recommend to move this paragraph to somewhere else in the paper. the same applies to paragraph 147-160. Author may crate new heading as immune response in MIS-C for these 2 paragraphs.

Re: following the suggestions, the paragraph has been splitted in different sections.

line 134: please explain which viral proteins are targeted by these antibodies 

Re: this information has been reported in the text

Figure: recommend adding NRF2 pathway to figure.

Re:: Figure 2 has been changed.

ACE1/ACE2 balance is very important as ACE1 is pro inflammatory and pro ROS and ACE2 is anti inflammatory and anti oxidant. Recommend that author talks about the importance of ACE2 pathway and if there is any studies on ACE 2 pathway in MIS-C patients.

Re: The link between ACE1/ACE2 and ROS has been better defined in the text. However, to the best of our knowledge, there are not studies investigating the role of ACE-2 in patients with MIS-C.

Is there any studies which looked at the oxidative stress markers with patients with MIS-C, pediatric or adult/MIS-A. 

Re: to the best of our knowledge, there are not studies with the primary outcome the investigation of  reliable oxidative stress markers in patients with MIS-C, pediatric or adult/MIS-A.

Author should also notice and discuss this dilemma. Although antioxidant capabilities of the children are much better than elderly MIS-C was initially described in children and  much more prevalent in children then elderly. If oxidative stress is very important in the pathogenesis of MIS-C then why we don't see much higher MIS-C/MIS-A cases in elderly with chronic disease like metabolic syndrome, DM, cardiovascular disease that has common factor of inflammation and elevated oxidative stress.

Re: We thank the reviewer for this consideration. Oxidative stress is an important hallmark in several diseases characterized by predisposition to alteration of inflammatory function and/or premature ageing. MIS-C was initially described in children and much more prevalent in children then adult (MIS-A). In aged individual a prooxidant state is also often associated with a mitochondrial dysfunction and a declines in expression and activity of endoplasmic reticulum molecular chaperones. A delicate balance between oxidants and antioxidants is essential for physiological functioning. On the contrary, the loss of this balance usually leads to dysfunctions and cellular damage at various levels, including membrane phospholipids, proteins, and nucleic acid. Although significantly less data exists for MIS-A, there is extensive clinical and laboratory overlap between the two conditions. By measuring redox biomarker profile, such as concentration of protein and lipid peroxidation products, the presence of a prooxidant state could be documented, both in children and in erderly. This concept has been added in the text

Reviewer 2 Report

The manuscript is an interesting narrative review on pediatric MIS-C. It is well-written and exhaustive. The figures and tables are informative. I only have a few minor comments:

-          In the title, I suggest adding “pediatric” (“pediatric multisystemic syndrome associated….”)

-          In the keywords section, you should delete the numbers 1, 2, and 3.

-          Please check the spelling of the Authors’ names.

-          I suggest implementing the reference list with the following: 10.1172/JCI144554;  10.3390/DIAGNOSTICS11091647; DOI:10.1177/0025817220938004; 10.1007/s42399-020-00690-6

Author Response

Reviewer 2:

The manuscript is an interesting narrative review on pediatric MIS-C. It is well-written and exhaustive. The figures and tables are informative. I only have a few minor comments:

-          In the title, I suggest adding “pediatric” (“pediatric multisystemic syndrome associated….”)
Re: The title has been changed according the suggestion

-          In the keywords section, you should delete the numbers 1, 2, and 3.

Re: The numbers have been deleted

-          Please check the spelling of the Authors’ names.

Re: The spelling of Author names have been checked

-          I suggest implementing the reference list with the following: 10.1172/JCI144554;  10.3390/DIAGNOSTICS11091647; DOI:10.1177/0025817220938004; 10.1007/s42399-020-00690-6

Re: The suggested reference has been added and discussed in the text

Reviewer 3 Report

Dear Authors,

thank you for trying to sum up all the information regarding MIS-C.

Here are some suggestions, because I miss some points.

1. The introduction should have a 1st paragraph with most common symptoms of SARS-CoV2 infection, including the following aspects:

- Martynowicz H, Jodkowska A, PorÄ™ba R, Mazur G, WiÄ™ckiewicz M. Demographic, clinical, laboratory, and genetic risk factors associated with COVID-19 severity in adults: A narrative review. Dent Med Probl. 2021;58(1):115–121. doi:10.17219/dmp/131795

- Paradowska-Stolarz AM. Oral manifestations of COVID-19 infection: Brief review. Dent Med Probl. 2021;58(1):123–126. doi:10.17219/dmp/131989

2. In table 1, please explain "pediatric age group" & delete numbers "1" before writing about the age group

3.Add a chapter and a graph on the inclusion and exclusion cryteria (Prisma flow diagram)

4. I would also add the table summing up all the abbreviations used in your article

4. Please, add the limitations of the paper.

Thank you in advance.

Author Response

Reviewer 3:

Dear Authors,

thank you for trying to sum up all the information regarding MIS-C.

Here are some suggestions, because I miss some points.

  1. The introduction should have a 1st paragraph with most common symptoms of SARS-CoV2 infection, including the following aspects:

- Martynowicz H, Jodkowska A, PorÄ™ba R, Mazur G, WiÄ™ckiewicz M. Demographic, clinical, laboratory, and genetic risk factors associated with COVID-19 severity in adults: A narrative review. Dent Med Probl. 2021;58(1):115–121. doi:10.17219/dmp/131795

Paradowska-Stolarz AM. Oral manifestations of COVID-19 infection: Brief review. Dent Med Probl. 2021;58(1):123–126. doi:10.17219/dmp/131989

Re: we added and comment in the text the suggested references

  1. In table 1, please explain "pediatric age group" & delete numbers "1" before writing about the age group

Re: Table 1 as been changed according the suggestion

3.Add a chapter and a graph on the inclusion and exclusion cryteria (Prisma flow diagram)

Re: Prisma flow diagram has been added in Figure 1

  1. I would also add the table summing up all the abbreviations used in your article

Re: a table with all abbreviation has been added

  1. Please, add the limitations of the paper.

Re: Limitations of the manuscript have been added

Round 2

Reviewer 3 Report

Dear Authors, thank you for the corrections. 

The one last small think, that you could correct on the phase of proofreading is: 

"List of abbreviations" - you should add "used in the article"

and that would be perfect. Thank you